# Phytocomplex of a Standardized Extract from Red Orange (*Citrus sinensis* L. Osbeck) against Photoaging

**DOI:** 10.3390/cells11091447

**Published:** 2022-04-25

**Authors:** Barbara Tomasello, Giuseppe Antonio Malfa, Rosaria Acquaviva, Alfonsina La Mantia, Claudia Di Giacomo

**Affiliations:** 1Department of Drug and Health Science, University of Catania, Viale A. Doria 6, 95125 Catania, Italy; btomase@unict.it (B.T.); alfy.lamantia@gmail.com (A.L.M.); cdigiaco@unict.it (C.D.G.); 2Research Centre on Nutraceuticals and Health Products (CERNUT), University of Catania, Viale A. Doria 6, 95125 Catania, Italy; 3Nacture S.r.l., Spin-Off University of Catania, 95125 Catania, Italy

**Keywords:** antioxidant activity, ROS, comet assay, flavonoids, cyanidin, collagen, elastin, TNF-α, IL-6, MMPs

## Abstract

Excessive exposure to solar radiation is associated with several deleterious effects on human skin. These effects vary from the occasional simple sunburn to conditions resulting from chronic exposure such as skin aging and cancers. Secondary metabolites from the plant kingdom, including phenolic compounds, show relevant photoprotective activities. In this study, we evaluated the potential photoprotective activity of a phytocomplex derived from three varieties of red orange (*Citrus sinensis* (L.) Osbeck). We used an in vitro model of skin photoaging on two human cell lines, evaluating the protective effects of the phytocomplex in the pathways involved in the response to damage induced by UVA-B. The antioxidant capacity of the extract was determined at the same time as evaluating its influence on the cellular redox state (ROS levels and total thiol groups). In addition, the potential protective action against DNA damage induced by UVA-B and the effects on mRNA and protein expression of collagen, elastin, MMP1, and MMP9 were investigated, including some inflammatory markers (TNF-α, IL-6, and total and phospho NFkB) by ELISA. The obtained results highlight the capacity of the extract to protect cells both from oxidative stress—preserving RSH (*p* < 0.05) content and reducing ROS (*p* < 0.01) levels—and from UVA-B-induced DNA damage. Furthermore, the phytocomplex is able to counteract harmful effects through the significant downregulation of proinflammatory markers (*p* < 0.05) and MMPs (*p* < 0.05) and by promoting the remodeling of the extracellular matrix through collagen and elastin expression. This allows the conclusion that red orange extract, with its strong antioxidant and photoprotective properties, represents a safe and effective option to prevent photoaging caused by UVA-B exposure.

## 1. Introduction

Skin is the organ with the largest contact area between the human body and the external environment against which it acts as a barrier.

It not only protects the body from external environmental damage and prevents water loss, but it also has a certain cosmetic effect. As the largest organ of the human body, the skin shows obvious signs of aging due to various causes including environmental pollutants. Ultimately, the events that characterize skin aging can be mainly classified into chronological aging events and photoaging events [1]. Chronological aging, due to advancing age, is strongly influenced by ethnicity, the individual, and the skin site. It generates internal changes in the skin due to a decrease in the functionality of stem cells with a consequent slowdown in the keratinization process and therefore in skin repair. Photoaging, on the other hand, is caused by prolonged sun exposure particularly to ultraviolet radiation (UVR) such as UVA and UVB. UVR can cause skin problems, and this is due to its ability to damage cells by producing free radicals, reactive oxygen species (ROS), and direct injuries to genetic material [2,3]. This occurs because ultraviolet rays, which are electromagnetic waves, carry energy capable of promoting the formation of reactive species and triggering chemical reactions. The extent of the damage increases as the energy transported increases, which is inversely proportional to the wavelength: the energy grows passing from UVA to UVB and up to UVC, which is strongly attenuated by the atmosphere and does not result in significant irradiation on Earth [4].

UVA radiation, characterized by a wavelength between 320 and 400 nm, has low energy but, being strongly penetrative, generates various effects on the dermis. In fact, these waves, promoting the production of the metalloprotease of the matrix, accelerate the hydrolysis of collagen and the consequent destruction of the tissue and progressive degeneration of the extracellular matrix of the dermis. Furthermore, UVA inhibits the synthesis of hyaluronic acid because it induces the downregulation of the enzyme hyaluronic acid synthetase [5,6,7,8].

UVB radiation (λ = 320 and 280 nm) has higher energy, but the reduced penetrating capacity does not allow it to reach the dermis. These waves act at the level of keratinocytes located in the epidermis layer and, in cases of frequent exposure, can generate DNA damage and keratinocyte mutations and release soluble cytokines that are important risk factors for aging, inflammation, apoptosis, and carcinogenesis. However, keratinocytes, unlike fibroblasts, are very resistant because they can counteract these harmful events through their defense system [9].

UVR-induced oxidative stress causes direct and indirect DNA damage. The direct damage occurs when the DNA absorbs the UV-B photon, resulting in a rearrangement in the nucleotide sequence and the consequent deletion or mutation of the DNA strand. The indirect damage is highlighted when the DNA absorbs UVA, promoting the transfer of electrons and energy to the oxygen molecules. This leads to the formation of singlet oxygen ions of free radicals, causing DNA damage [2].

It is known that a balanced diet, rich in fruit and vegetables, and a healthy lifestyle play fundamental roles in counteracting skin aging [10]. Nutrition is closely associated with skin health and is crucial for all the biological processes of the skin, including aging or disease. Nutritional levels and eating habits can repair damaged skin but can also cause damage in turn. Furthermore, a good deal of recent research has demonstrated the photoprotective activities of several phytochemicals [11].

Data from the literature report that red oranges, *Citrus sinensis* (L.) Osbeck (Rutaceae), are a rich source of bioactive compounds such as polyphenols, flavonoids, and anthocyanins including ascorbic acid that have been shown to be effective in protecting the body from damage caused by ROS, in countering inflammatory processes, and in reducing the risk of metabolic disorders and chronic diseases such as cancer [12,13,14,15,16].

In this study, we evaluated the effects of a standardized extract obtained from the juice of three red orange varieties (*Citrus sinensis* (L.) Osbeck)—Moro, Sanguinello, and Tarocco—on an experimental in vitro model of skin photoaging in order to verify the hypothesis that the natural antioxidants present in the extract may protect and prevent UV-induced skin damage. Therefore, to understand the mechanism of action of the standardized extract, in this study, we evaluated its effects on the main biological markers of photoaging on two cell lines: human foreskin fibroblasts (HFF-1) and immortalized human keratinocytes (NCTC 2544) irradiated with different doses of UVA and UVB such as to simulate a solar exposure unable to induce sunburn.

## 2. Materials and Methods

### 2.1. Chemicals

All solvents, chemicals, and reference compounds were purchased from Sigma-Aldrich (Milan, Italy) except when otherwise specified.

### 2.2. Extract Specifications

The Red Orange Complex H extract (ROC H^®^) was obtained from the juice of three red orange (*Citrus sinensis* (L.) Osbeck) varieties: Moro, Sanguinello, and Tarocco with batch number 03202004-01. It was provided by Bionap S.R.L. (Belpasso, Catania, Italy) and standardized to the content of the following bioactive constituents: ferulic acid (2.2% *w*/*w*), narirutin (0.4% *w*/*w*), hesperidin (9.0% *w*/*w*), total flavanones (9.4% *w*/*w*), cyanidin 3-O-glucoside (3.1% *w*/*w*), and ascorbic acid (6.5% *w*/*w*). Quantification and identification of the secondary metabolites were conducted using HPLC-DAD analysis, except for cyanidin-3-O-glucoside quantified by UV spectroscopy. The extract was authenticated by DNA barcoding, and molecular diagnostic tests were performed by TRU-ID Ltd. (TRU-ID Ltd. Research, Guelph, ON, Canada). The certificate of authentication, including further product specification on purity, is available in the Appendix A Section.

### 2.3. SOD-Like Activity and DPPH Antioxidant Assay

The scavenger effect of *C. sinensis* extract (0.08–0.160–0.250–0.320 µg/mL) on superoxide anion (SOD-like activity) was recorded as a decrease in absorbance at λ = 340 nm, as previously reported by Malfa et al. [17]. Results are expressed as the percentage of inhibition of NADH oxidation; as a reference compound, superoxide dismutase (SOD) (80 mU) was used. Results represent the average ± S.D. of three independent experiments. The free radical scavenging capacity, analyzed through the ability of the extract to bleach stable 1,1-diphenyl-2-picryl-hydrazyl radical (DPPH), was compared to Trolox (30 µM), a water-soluble derivative of vitamin E, used as a reference compound. After 10 min at room temperature, the absorbance at λ = 517 nm of the DPPH reaction mixture containing different concentrations of *C. sinensis* extract (20–40–80–160 µg/mL) in 1 mL of ethanol was recorded [15]. The results were obtained from the average of three independent experiments and are expressed as the % decrease in absorbance ± S.D.

### 2.4. Cell Culture and Treatments

Human foreskin fibroblasts (HFF-1) were obtained from the American Type Culture Collection (ATCC, Rockville, MD, USA). HFF-1 were cultured in Dulbecco’s Modified Eagle’s Medium supplemented with 15% fetal bovine serum, 4.5 g/L glucose, 100 U/mL penicillin, and 100 mg/mL streptomycin. Immortalized human keratinocytes (NCTC 2544), isolated from the epidermis, were provided by Interlab Cell Line Collection, Genoa, Italy. NCTC 2544 were cultured in MEM medium (Minimal Essential Medium) containing 10% fetal serum and 1% penicillin/streptomycin in a 5% CO_2_ atmosphere at 37 °C. After 24 h of incubation in a humidified atmosphere of 5% CO_2_ at 37 °C to allow cell attachment, the cells were treated for 4 h with different concentrations (0.1–1–10–50–100–200–400 µg/mL) of *C. sinensis* extract previously dissolved in the medium and subsequently irradiated with UVA and/or UVB at different doses and times.

### 2.5. Cell Culture Irradiation with UVA and UVB

The irradiations were performed using 2 different types of photoreactors. For the UVA, the Rayonet was used equipped with two PRP lamps with emission centered at λ = 350 nm (UVA) and λ = 300 nm (UVA) (~1 mW/cm^2^), while for the UVB, the photoreactor was used with reflecting wall equipped with a lamp with emission centered at λ = 280–300 nm (UVB) (~2 mW/cm^2^) [18]. The doses of UVA and UVB used for the experiments were those capable of simulating sun exposure. Irradiation doses were calculated using the formula below: irradiation dose (mJ/cm^2^) = exposure time (s) × irradiance (mW/cm^2^). NCTC 2544 and HFF-1 cells were irradiated with UVA at different doses (1, 5, 10, and 15 J/cm^2^) and with UVB at 5, 10, 20, 25, and 50 mJ/cm^2^. Five independently repeated experiments were performed to evaluate the effect of irradiation and combined treatment with *C. sinensis* extract. For all other experiments, the UVA dose used for irradiation of NCTC 2544 and HFF-1 cells was 15 J/cm^2^, while that of UVB was 25 and 50 mJ/cm^2^.

### 2.6. MTT Bioassay

Cell viability was performed on a 96 multiwell plate (8 × 10^3^ cells/well) using MTT assay that measures the conversion of tetrazolium salt to yield colored formazan in the presence of metabolic activity. The amount of formazan is proportional to the number of living cells [15]. The optical density was measured with a microplate spectrophotometer reader (Titertek Multiskan, Flow Laboratories, Helsinki, Finland) at λ = 570 nm. Results are expressed as percentage cell viability with respect to control (untreated cells).

### 2.7. Total Thiol Group Determination

Total thiol groups (RSH) were measured using a spectrophotometric assay that was done following the procedure previously described [19]. Briefly, 200 µL of lysate supernatant was added to 2,2-dithio-bis-nitrobenzoic acid (DTNB), and the reaction of thiols with DTNB was measured at λ = 412 nm. Results are expressed as µmol/mg protein.

### 2.8. Reactive Oxygen Species Assay

ROS levels were measured by dichlorofluorescein diacetate (DCFH-DA) assay as previously described [20]. The fluorescence intensity was detected by fluorescence spectrophotometry (excitation, λ = 488 nm; emission, λ = 525 nm). Results are expressed as fluorescence intensity/mg protein, and for each sample, the total protein content was determined using the Sinergy HTBiotech instrument by measuring the absorbance difference at λ = 280 and λ = 260.

### 2.9. Comet Assay

HFF1 and NCTC 2544s cells (1.5 × 10^5^ cells/mL) were pretreated with *C. sinensis* extract (50 and 100 μg/mL) followed by UVA or UVB irradiation in 6-well plates. After treatments, cells embedded with 0.7% low melting point agarose (LMA) were spotted on pre-treated FlareTM slide (Trevigen Inc., Gaithersburg, MD, USA) and incubated in cold lysis solution (NaCl 2.5 M, Na_2_EDTA 100 mM, TRIS-HCl 8 mM, TRITON 1%, and DMSO 10%, pH 10) for 1 h at 4 °C. The nucleoids were electrophoresed for 20 min at 0.7 V/cm in alkaline solution (1 mM Na_2_EDTA and 300 mM NaOH), washed with neutralization buffer, dried with 70% ethanol, and stained with SYBR Green (1:10,000). Fifty nucleoids were analyzed for each sample using an epifluorescence microscope (Leica, Wetzlar, Germany) equipped with a camera. CASP (1.2.2, CASPLab, University of Wroclaw, Wroclaw, Poland) image analysis software was used to evaluate DNA damage. The results are expressed as the percentage of fragmented DNA present in the comet tail (%TDNA) [21,22].

### 2.10. RNA Extraction and Reverse Transcription-Polymerase Chain Reaction (RT-PCR)

HFF1 and NCTC 2544 cells (1.5 × 10^5^ cells/mL) were pretreated with *C. sinensis* extract (25 and 50 μg/mL) followed by UVB irradiation in 6-well plates and further incubated for 24 h. Total RNA was isolated using TRIzol (Invitrogen, Carlsbad, CA, USA) following the manufacturer’s instructions. A total of 1 µg of total RNA was used to prepare the complementary DNA (cDNA) using QuantiTect Reverse Transcription Kit (Qiagen Inc., Valencia, MD, USA). RT-PCR were performed with the QuantiNova SYBR Green RT-PCR Kit according to the manufacturer’s instructions on Rotor-Gene Q5PLEX using various predesigned and bioinformatically validated primers sequences (QuantiTect Primer Assays, Qiagen Inc., Valencia, MD, USA) for COLA1A1, ELN, MMP1, and MPP9, according to previously described methods [15]. mRNA expression levels were normalized to GAPDH (glyceraldehyde-3-phosphate dehydrogenase), and all experiments were repeated three times.

### 2.11. Western Blotting Analysis

The lysates of HFF1 and NCTC-2544 cells (3 × 10^5^ cells/mL) were prepared using radioimmunoprecipitation assay (RIPA) buffer containing phosphatase and protease inhibitor cocktail (Sigma-Aldrich, St. Louis, MO, USA). The Bradford method was applied for the quantification of the protein content. Western blot analysis was carried out according to our previously described methods [22] using the following primary antibodies: anti-COL1A1 (Cat#91144, 1:1000 dilution), anti-MMP9 (Cat#3852, 1:1000 dilution), anti-MMP1(Cat#54376, 1:1000 dilution) (Cell Signaling Technology, Denvers, MA, USA), anti-ELN (SAB1405754 from Sigma-Aldrich, 1:1000 dilution), and anti-GAPDH (Millipore, Darmstadt, Germany, dilution 1:5000). The secondary goat antirabbit labeled with IRDye 680 (1:15,000) and goat antimouse labeled with IRDye 800 (1: 20,000) were used to detect primary antibodies. The Odyssey Infrared Imaging System (LI-COR Biosciences, Lincoln, NE, USA) was used to scan the blot, and quantitative densitometric analysis was performed by using ImageJ (http://imagej.nih.gov/ij/, accessed on 6 September 2021). The results are expressed as arbitrary densitometric units (A.D.U.) and were normalized with respect to GAPDH levels.

### 2.12. Determination of TNF-α, IL-6, and Total and Phospho NFkB p65

Commercially available enzyme-linked immunosorbent assay (ELISA) kits were used to measure the levels of TNF-α (BMS223HS, Invitrogen, Carlsbad, CA, USA) and IL-6 (BMS213HS, Invitrogen, Invitrogen, Carlsbad, CA, USA) in the cell culture supernatants and the level of total and phospho NFkB p65 (85-86083-11, Invitrogen) in cell lysate, according to manufacturer’s protocol. Absorbance of samples was read at λ = 450 nm by microplate reader (Synergy HT multi-mode microplate reader, BioTek, Milan, Italy). The TNF-α and IL-6 concentrations were calculated from a standard curve, while the average value of optical density (O.D.) at λ = 450 nm was used to evaluate the level of total and phospho NFkB p65. Results are reported as O.D. ratio of the phospho NFkB-sample O.D. value to the total NFkB-sample O.D. value.

### 2.13. Statistical Analysis

The experimental data are presented as the mean ± standard deviation (S.D.). The statistical analysis of the data was carried out either by one-way analysis of variance (ANOVA) or Kruskal–Wallis test depending on the results of the normality analysis (Shapiro–Wilk normality test) followed by Tukey’s post hoc test using the Graph Prism version 5. Differences were considered significant when *p* ≤ 0.05.

## 3. Results

### 3.1. Effects of C. sinensis Extract on HFF-1 and NCTC 2544 Cell Viability

The treatment of HFF-1 and NCTC 2544 with *C. sinensis* extract (0.1–1–10–50–100–200–400 μg/mL) for 24 h and 5 days was unable to affect cell viability (Appendix A). Since *C. sinensis* extract did not induce toxicity, in this research, 100 μg/mL was used as the maximum concentration of extract in order to avoid unnecessarily high concentrations.

### 3.2. Effects of UVA and UVB Irradiation on Cell Viability of HFF-1 and NCTC 2544 Pretreated with C. sinensis Extract

HFF-1 and NCTC 2544 cells were irradiated with UVA at 1, 5, 10, and 15 J/cm^2^ and with UVB at 5, 10, 20, 25, or 50 mJ/cm^2^ (Figure 1A–D). Cell viability was determined 24 h after irradiation. In HFF-1 and NCTC 2544 cells, irradiation with UVA at a dose of 1 J/cm^2^ did not modify cell viability, unlike doses of 5, 10, and 15 J/cm^2^ that significantly decreased it in a dose-dependent manner (Figure 1A,B). The dose of 15 J/cm^2^ induced approximately 35% and 65% of cell viability in HFF1 and NCTC 2544 cells, respectively. Figure 1A,B show that keratinocytes were much more resistant than fibroblasts to UVA radiation. Figure 1C shows that cell viability of UVB-irradiated fibroblasts decreased in a dose-dependent manner, reaching approximately 70% and 60% at doses of 25 and 50 mJ/cm^2^, respectively. UVB irradiation of NCTC 2544 induced a reduction in cell viability in a dose-dependent manner, reaching 40% and 30% at 25 mJ/cm^2^ and 50 mJ/cm^2^, respectively (Figure 1D).

Pretreatment with *C. sinensis* extract (0.5–5–50–100 µg/mL) for 4 h was able to increase in a dose-dependent manner cell viability of HFF-1 irradiated with a dose equal to 15 J/cm^2^ of UVA (Figure 2A,B). At concentrations of 5, 50, and 100 µg/mL, the extract protected HFF-1 and NCTC 2544 cells from radiation-induced damages by increasing their cell viability in a dose-dependent manner (Figure 2A,B). Figure 2B confirms that keratinocytes are much more resistant than HFF-1 cells to UVA irradiation, and pretreatment with *C. sinensis* extract slightly increased vitality compared to untreated irradiated cells.

Cell viability of HFF-1, reduced by UVB irradiation at 25 mJ/cm^2^ and 50 mJ/cm^2^, was increased by pretreatment with the *C. sinensis* extract (Figure 3). In particular, the protective effect was greater in cells irradiated at 50 mJ/cm^2^ (Figure 3A,B). Figure 3C,D confirm that the cell viability of NCTC 2544 irradiated at 25 and 50 mJ/cm^2^ decreased by about 60% compared to the control and that the pretreatment with the extract (50 and 100 μg/mL) was able to increase viability in a dose-dependent manner with respect to irradiated and untreated cells.

### 3.3. Cell-Free Antioxidant Properties of C. sinensis Extract

#### 3.3.1. SOD-Like Activity

The scavenger activity of the extract was determined using a method that excluded Fenton-type reactions and the xanthine/xanthine oxidase system. The extract was found to have a significant, dose-dependent scavenger effect, and its effect at 0.320 μg/mL was comparable to that performed by the reference control superoxide dismutase (SOD 80 mU) (Figure 4A).

#### 3.3.2. DPPH Radical Scavenging Activity Assay

The scavenger activity of the extract was also confirmed by testing its quenching effect using the DPPH test. The extract showed a significant dose-dependent quenching effect reaching an efficacy of almost 90% at a concentration of 160 μg/mL; this effect was comparable to that of 30 μM Trolox (Figure 4B). Therefore, the results from this assay demonstrate that the quenching activity of *C. sinensis* extract was obtained at concentrations higher than those on the superoxide anion. This is probably due to the steric hindrance carried out by the radical itself and, therefore, to the smaller size of the O_2_ radical compared to the DPPH radical. In fact, it is known that the steric accessibility of the DPPH radical represents the determining factor for the reaction; thus, small molecules, with better access to the radical site, show a higher antioxidant power.

### 3.4. In Cell Antioxidant Properties of C. sinensis Extract

#### 3.4.1. *C. sinensis* Extract Preserved Total Thiol Group Levels

To clarify the protective mechanism of the *C. sinensis* extract, we measured the levels of RSH in HFF-1 and NCTC 2544 cells pretreated for 4 h with 50 μg/mL and 100 μg/mL of extract and subsequently irradiated with UVA and/or UVB. The determinations were carried out 3 h after irradiation, in order to assess the cell response properly. In HFF-1 and NCTC 2544 cells, the irradiation with UVA at 15 mJ/cm^2^ and with UVB at 25 mJ/cm^2^ or 50 mJ/cm^2^ was able to induce a decrease in RSH levels compared to the nonirradiated controls. In HFF-1 cells, the treatment with 50 and 100 μg/mL in irradiated cells preserved RSH levels and counteracted the decrease in RSH levels (Figure 5A). On the other hand, the *C. sinensis* extract was able to bring the levels of RSH back to control values in NCTC 2544 cells (Figure 5B). UVB irradiation, by reducing RSH levels, makes fibroblasts sensitive to cell damage and therefore susceptible to cell death. The results obtained confirm that *C. sinensis*, significantly counteracting the decrease in RSH levels, may protect irradiated cells (Figure 5C). A significant decrease in RSH levels was also detected in the NCTC 2544 cells irradiated with UVB and, consistent with the results obtained in the MTT assay, the pretreatment with the extract, although it countered the reduction, was not able to restore control RSH levels (Figure 5D).

#### 3.4.2. *C. sinensis* Extract Reduced Reactive Oxygen Species Levels

To define the role of oxidative stress in photoaging, we measured the ROS levels in HFF-1 and NCTC 2544 irradiated with UVA and UVB. The irradiation resulted in increased ROS levels in both cell lines. Consistent with the results of RSH levels, pretreatment with *C. sinensis* extract reduced ROS levels both in HFF-1 and NCTC 2544 cells irradiated with UVA and UVB, confirming the antioxidant ability of the extract (Figure 6).

### 3.5. C. sinensis Extract Counteracted DNA Damage Induced by UVA and UVB Irradiation

To confirm the protective effect of the extract, DNA damage was also evaluated in cells using COMET assay. UVA and UVB irradiation in both cell lines resulted in DNA damage that was counteracted by pretreatment with *C. sinensis* extract in a dose-dependent manner (Figure 7).

### 3.6. Protective Effects of C. sinensis Extract on Extracellular Matrix Remodeling: Collagen Type I, Elastin, MMP-1, and MMP-9

We examined whether the *C. sinensis* extract can modulate the ECM remodeling biomarkers, by evaluating the transcriptional and translational expression of elastin (ELN), type I collagen (COL1A1), and Matrix Metalloproteinases (MMP1 and MPP9) by RT-PCR and immunoblotting assay. As shown in Figure 8 for HFF-1 cells, UVB treatment markedly downregulated the transcriptional and translational level of type I collagen and, to a lesser extent, that of elastin, while the pretreatment with *C. sinensis* extract restored their expressions. Unlike HFF-1 cells, no expression of type I collagen and elastin was detected in human NCTC 2544 keratinocyte cells (Figure 9D). Moreover, UVB exposure induced the expression of MMP9 and MMP1 in both cell lines, and pretreatment with *C. sinensis* extract significantly reduced the UVB-stimulated MMP mRNA expression and protein levels, in a concentration-dependent fashion (Figure 8B,E–G and Figure 9A–D).

### 3.7. Positive Effects of C. sinensis Extract on the Expression of Inflammatory Markers: TNF-α, IL-6, and NFκB

To verify the involvement of inflammation in photoaging, medium from unirradiated or UVB-irradiated cells, with or without *C. sinensis* extract pretreatment, was collected 24 h post irradiation and assayed for TNF-α, IL-6, and total and phospho-NFkB by ELISA. The TNF-α was undetectable in unirradiated fibroblasts and keratinocytes, while the irradiation enhanced the TNF-α release in a dose-dependent manner in both cell lines. This UVB effect was especially evident in human keratinocytes. By *C. sinensis* extract pre-treatment, TNF-α levels were reduced, and the 50 µg/mL *C. sinensis* extract concentration was the most efficient with a drop of nearly 50% (Figure 10A,B). Upon UVB irradiation, IL-6 showed a similar trend to TNF-α with a drastic increase in IL-6 release. Only a pre-treatment with 50 µg/mL of *C. sinensis* extract decreased the IL-6 amount about 35–40% in both cell lines exposed to 25 and 50 mJ/cm^2^ UVB (Figure 10C,D). The activation of NFκB p65 by phosphorylation was significantly raised by UVB but inhibited/lessened by *C. sinensis* extract in a concentration-dependent manner (Figure 10E–H).

## 4. Discussion

Among the different types of oranges, the varietal group of red oranges has the particularity of containing a high quantity of polyphenols, flavonoids, ascorbic acid, and anthocyanins, both in the pulp and in the epicarp [16]. Many experimental results have shown that red orange extracts demonstrate a potent antioxidant activity, anti-inflammatory properties, and cytoprotective effects thereby preventing different chronic pathological conditions [23,24,25,26,27,28]. Recently, it was reported that polyphenols may act as a sunscreen from within, by reducing inflammation and oxidative stress and by protecting DNA from the harmful effects of solar radiation [29]. Photoaging is mainly due to exposure to UVA and UVB radiation whose main features and by different molecular mechanisms, alter cellular redox state, DNA integrity, and skin extracellular matrix homeostasis and induce proinflammatory processes [30].

One of the causes of photoaging is certainly the oxidative stress induced by ultraviolet radiation that can cause a decrease in the levels of endogenous antioxidants present in the skin [31,32,33]. The protective effects of plant extracts are closely related to their active ingredient content; in fact, there is a strong correlation between polyphenol content and the antioxidant power of a drug [12]. Results from cell-free antioxidant assays (Figure 4) confirm that the phytocomplex present in the extract composed of hydroxycinnamic acids, flavanones, anthocyanins, and vitamin C have significant antioxidant properties, capable of counteracting radical chain reactions and preventing possible damage from oxidative stress. The antioxidant effects were validated in both cell lines irradiated with different doses of UVA and UVB. The increased levels of ROS due to radiation exposure were neutralized by the extract, which, at a concentration of 100 µg/mL, was able to maintain ROS levels comparable to those measured for control cells (Figure 6). In addition, the RSH content, the most important intracellular redox buffer with glutathione as the main representative nonprotein thiol [34], was completely restored by the pretreatment at 100 µg/mL so maintaining the cellular environment reduced (Figure 5). Low intracellular nonprotein thiol levels are frequently considered a biomarker of aging as well as of pathological conditions [35]. The prevention of oxidative damage induced by UVR in the skin is strictly associated with DNA integrity. In fact, UVR affects DNA directly via photon absorption and indirectly by ROS production, either way inducing genotoxic effects [36]. Among the secondary metabolites of red oranges, flavonoids are a main component, capable of genoprotective activities directly related to their antioxidant action [37]. In fact, flavonoids and other secondary metabolites exerted a protective action in a dose-dependent manner in both cell lines irradiated with UVA-B, as demonstrated by the results of the comet assay. This further confirms the positive result of the pretreatment with the extract on the redox state and on the integrity of the DNA (Figure 7). According to other studies, in fact, the prevention of oxidative damage in the skin through a balanced diet and the exogenous integration of antioxidants, could provide valid help in increasing the endogenous skin protection systems, representing an effective strategy for the protection of the skin from oxidative damage mediated by UVR [31,32,38].

UVR acts on the dermis of the skin, accelerating the hydrolysis of skin collagen and promoting the production of matrix metalloproteinases (MMPs), responsible for the destruction of tissues and the progressive degeneration of the extracellular dermal matrix [6,7,39]. In our experimental model, the pretreatment with *C. sinensis* extract was able to reduce the deleterious effects of UVB radiation by increasing mRNA and protein expression of type 1 collagen and elastin in HFF1 cells and concomitantly decreasing, in both cell lines, the MMP1 and MMP9 protein and mRNA expression levels with respect to irradiated control cells (Figure 8 and Figure 9). Previous studies demonstrated that in addition to vitamin C, polyphenols, particularly flavonoids, also increased collagen and elastin content and inhibit MMP expression in cell cultures [11,40].

The resultant oxidative stress induced by UVR with the generation of ROS, induces proinflammatory factors by altering the NFκB signal transduction pathways. Therefore, activated skin cells release soluble cytokines that promote the process of aging, inflammation, apoptosis, and carcinogenesis [10,41]. Red orange extracts showed marked anti-inflammatory effects both in vitro and in vivo [16,24]. Moreover, hesperidin, narirutin, and cyanidin were demonstrated to suppress proinflammatory cytokine genes including TNF-α, IL-1β, and IL-6 [24,42]. Reported results also show that pretreatment with *C. sinensis* extract limited the release of inflammatory mediators such as TNFα and IL-6 after exposure to UVB radiation (Figure 10), thus limiting the inflammatory cellular response in both cell lines. These results further confirm the antioxidant and anti-inflammatory properties of the extract. Moreover, the polyphenol content of the extract was also able to counteract the activation of NFκB p65 by phosphorylation, as demonstrated by the obtained values regarding the ratio between NFkB and p-NFkB (Figure 10). It has been shown that several flavonoids modulate the inflammation pathway by inhibiting NFkB phosphorylation and preventing its translocation into the nucleus, which is essential for the transcription of proinflammatory mediators responsible for long-lasting inflammatory processes [43]. The results obtained in the present study further confirm that by reducing phosphorylation of NFκB, *C. sinensis* extract may prevent and/or counteract the activation of the inflammation pathway, thus contributing to the inhibition of photoaging.

## 5. Conclusions

The present results demonstrate that phytochemicals contained in red orange standardized extract, mainly represented by flavanones and anthocyanins, exerted a photoprotective action on fibroblasts and keratinocytes exposed to UVA-B radiation, by preventing oxidative stress, DNA damage, extracellular matrix degradation, and remodeling and inflammatory responses related to photoaging. In fact, the phytocomplex confers photoprotection through the inhibition of cellular oxidative damage counteracting skin injuries. According to recent literature data in the field, the use of botanical extracts appears to be an effective additional strategy to reduce the incidence of UV-mediated oxidative photodamage.

Certainly, further preclinical and clinical research should be carried out to better confirm the photoprotective potential and the effectiveness of the oral administration of nutraceutical doses of the *C. sinensis* standardized extract.

## Figures and Tables

**Figure 1 cells-11-01447-f001:**
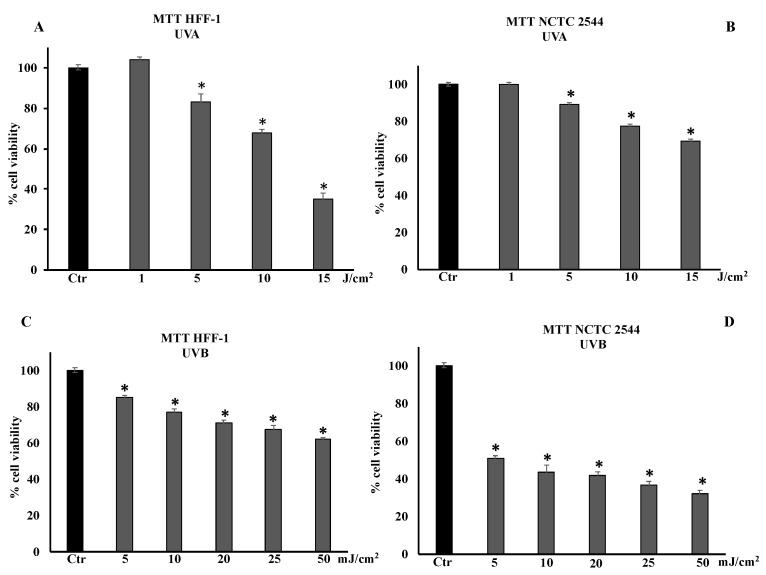
Cell viability in HFF-1 and NCTC 2544 non-irradiated (Ctr) and irradiated with UVA (**A**,**B**) and UVB (**C**,**D**) at different doses. Values are the mean ± S.D. of five experiments in triplicate. * Significant vs. untreated control cells: *p* < 0.001.

**Figure 2 cells-11-01447-f002:**
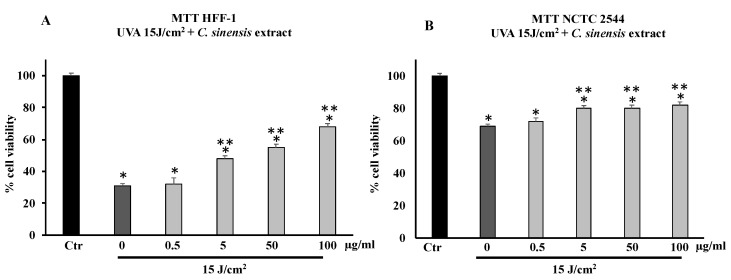
Cell viability in HFF-1 (**A**) and NCTC 2544 (**B**) cells untreated (Ctr) and treated for 4 h with *C. sinensis* extract at different concentrations (0.5–5–50–100 μg/mL) and irradiated with 15 J/cm^2^ of UVA. Values are the mean ± S.D. of five experiments in triplicate. * Significant vs. untreated control cells: *p* < 0.001; ** Significant vs. untreated irradiated control cells: *p* < 0.001.

**Figure 3 cells-11-01447-f003:**
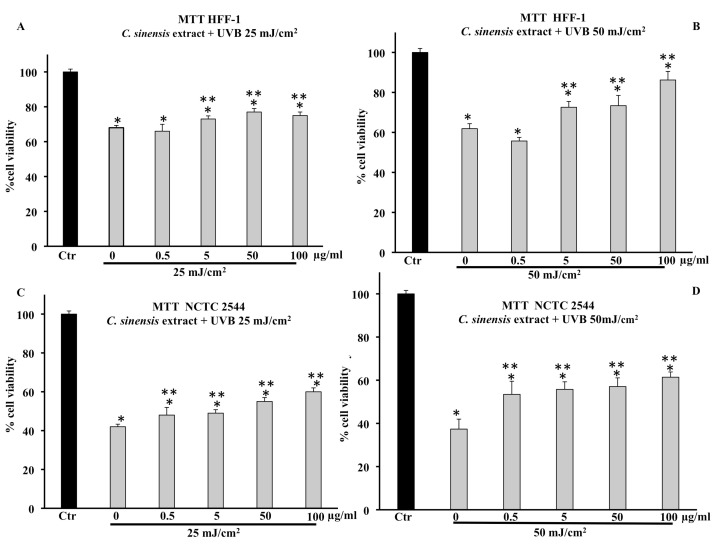
Cell viability in HFF-1 and NCTC 2544 cells untreated (Ctr) or treated for 4 h with *C. sinensis* extract at different concentrations (0.5–5–50–100 μg/mL) and irradiated with 25 mJ/cm^2^ (**A**,**C**) and 50 mJ/cm^2^ of UVB (**B**,**D**). Values are the mean ± S.D. of five experiments in triplicate. * Significant vs. untreated control cells: *p* < 0.001; ** Significant vs. untreated irradiated cells: *p* < 0.001.

**Figure 4 cells-11-01447-f004:**
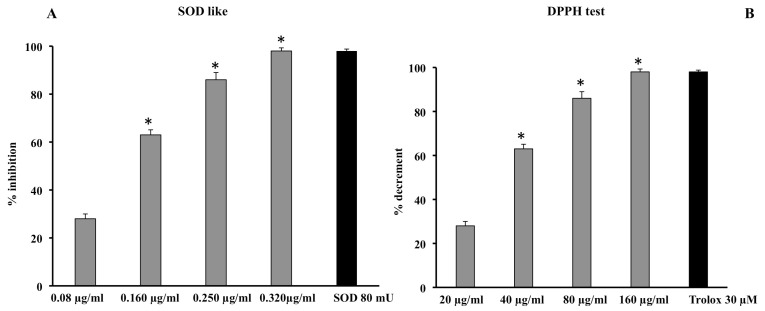
Scavenger effect of *C. sinensis* extract by SOD-like (**A**) and DPPH assay (**B**). Superoxide anion results (**A**) are expressed as percentage of inhibition of NADH oxidation (rate of superoxide anion production was 4 nmol/min). Quenching effect (**B**) results are expressed as percentage of the decrease in absorbance at λ = 517 nm when compared with the control. Each value represents the mean ± S.D. of three independent experiments in triplicate. * Significant vs. *p* < 0.001 vs. previous concentration.

**Figure 5 cells-11-01447-f005:**
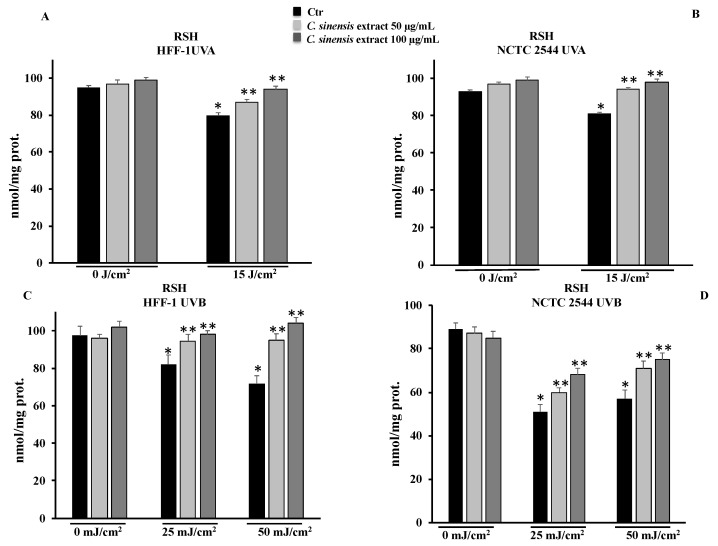
Total thiol groups in HFF-1 (**A**,**C**) and NCTC 2544 (**B**,**D**) cells untreated (Ctr) and treated for 4 h with *C. sinensis* extract at different concentrations (50–100 μg/mL) and irradiated with 15 J/cm^2^ of UVA and/or with 25 mJ/cm^2^ or 50 mJ/cm^2^ of UVB. Values are the mean ± S.D. of four experiments in triplicate. * Significant vs. untreated control cells: *p* < 0.001; ** Significant vs. untreated irradiated cells: *p* < 0.001.

**Figure 6 cells-11-01447-f006:**
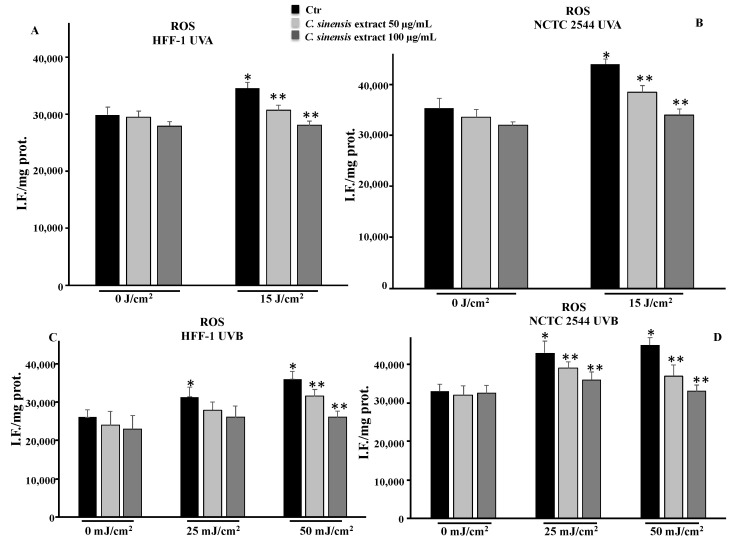
ROS levels in HFF-1 (**A**,**C**) and NCTC 2544 (**B**,**D**) cells untreated (Ctr) and treated for 4 h with *C. sinensis* extract at different concentrations (50–100 μg/mL) and irradiated with 15 J/cm^2^ of UVA and/or with 25 mJ/cm^2^ or 50 mJ/cm^2^ of UVB. Values are the mean ± S.D. of four experiments in triplicate. * Significant vs. untreated control cells: *p* < 0.001; ** Significant vs. untreated irradiated cells: *p* < 0.001.

**Figure 7 cells-11-01447-f007:**
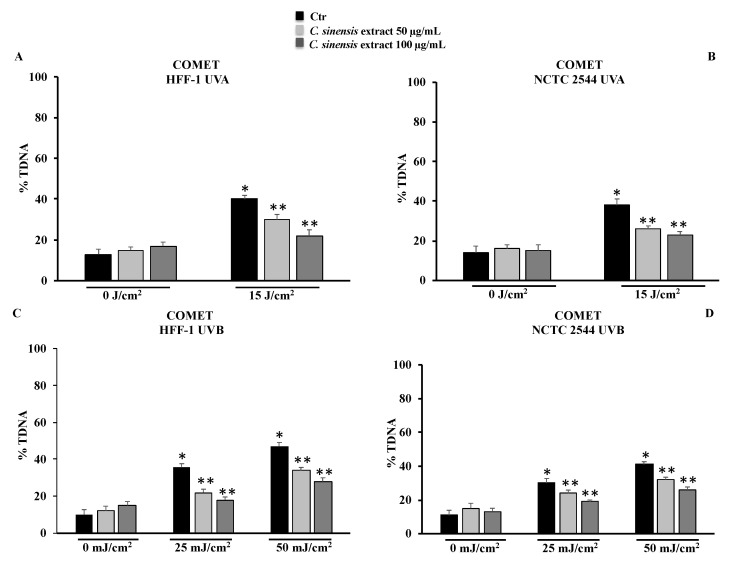
DNA damage was evaluated by alkaline comet assay in HFF-1 and in NCTC 2544 cells untreated and treated for 4 h with *C. sinensis* extract at different concentrations (50–100 μg/mL) and irradiated with UVA (**A**,**B**) and/or UVB (**C**,**D**). Results are expressed as percentage of DNA present in the comet tail (%TDNA). The data are reported as the mean ± S.D. from triplicate experiments. * Significant vs. untreated control cells: *p* ≤ 0.001; ** Significant vs. untreated irradiated cells: *p* ≤ 0.05.

**Figure 8 cells-11-01447-f008:**
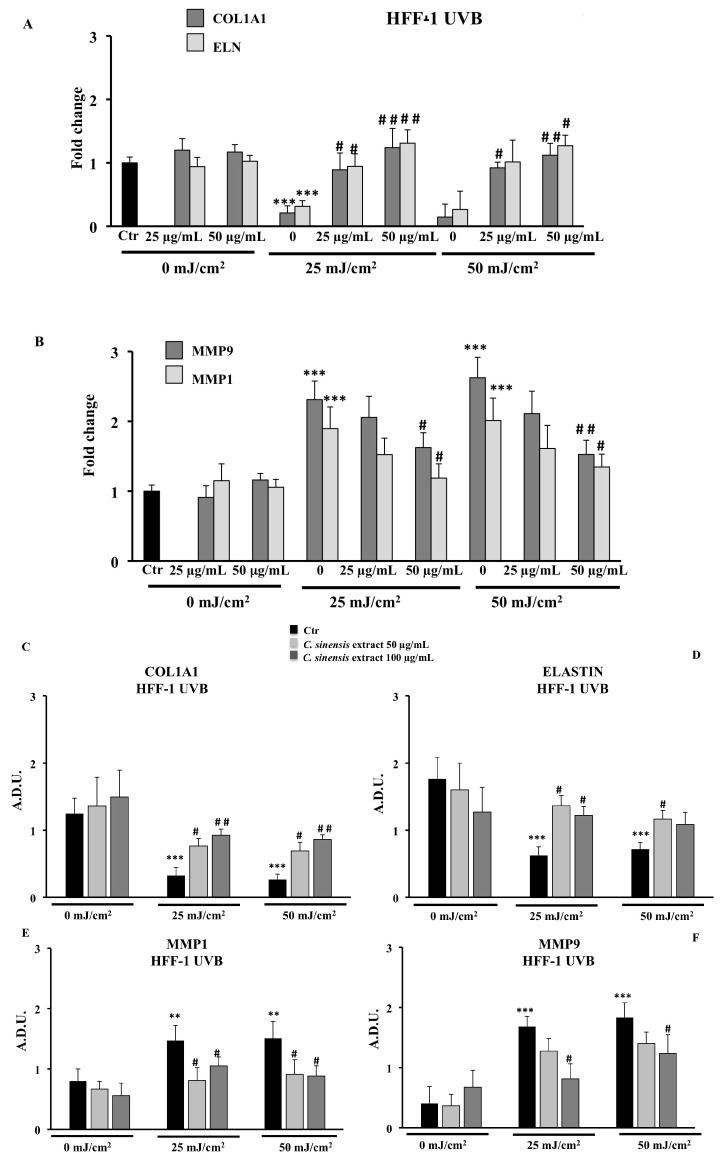
Effects of *C. sinensis* extract pretreatment (25 and 50 μg/mL) for 4 h on the regulation of (**A**) mRNA expression of type 1 collagen (COL1A1) and elastin (ELN) and (**B**) mRNA expression of MMP9 and MMP1; protein levels of COL1A1 (**C**), ELN (**D**), MMP1 (**E**), and MPP9 (**F**) after exposure to UVB (25 and 50 mJ/cm^2^) in HFF1 cells (**G**). Representative immunoblotting image of each protein examined. The data are reported as the mean ± S.D. from two experiments performed in quadruplicates. Dashes to the right of the membrane bands show positions of prestained molecular mass markers. ** Significant vs. untreated control cells: *p* < 0.01; *** Significant vs. untreated control cells: *p* < 0.001; ^#^ Significant vs. untreated irradiated cells: *p* ≤ 0.05; ^##^ Significant vs. untreated irradiated cells: *p* < 0.01.

**Figure 9 cells-11-01447-f009:**
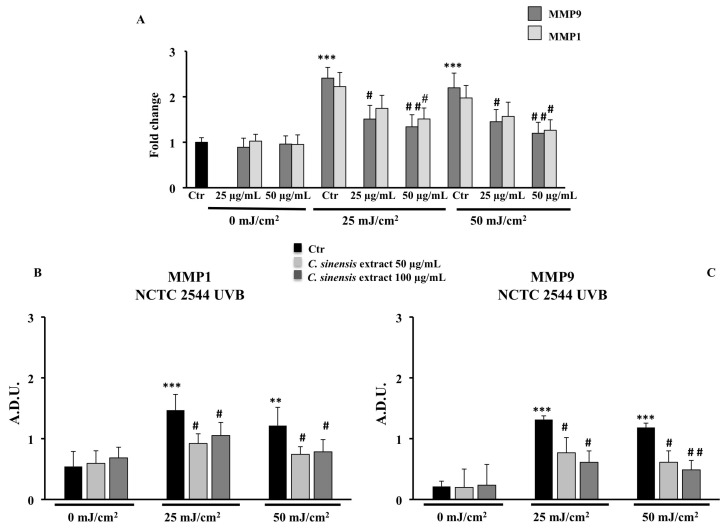
Effects of *C. sinensis* extract pretreatment (25 and 50 μg/mL) for 4 h on the regulation of (**A**) the mRNA expression of MMP1 and MMP9; (**B**,**C**) protein levels of MMP1 and MMP9 after exposure to UVB (25 and 50 mJ/cm^2^) in NCTC 2544 cells. (**D**) Representative immunoblotting image of each protein examined. The data are reported as the mean ± S.D. from two experiments performed in quadruplicates. Dashes to the right of the membrane bands show positions of prestained molecular mass marker. ** Significant vs. untreated control cells: *p* < 0.01; *** Significant vs. untreated control cells: *p* < 0.001; ^#^ Significant vs. untreated irradiated cells: *p <* 0.05; ^##^ Significant vs. untreated irradiated cells: *p* < 0.01.

**Figure 10 cells-11-01447-f010:**
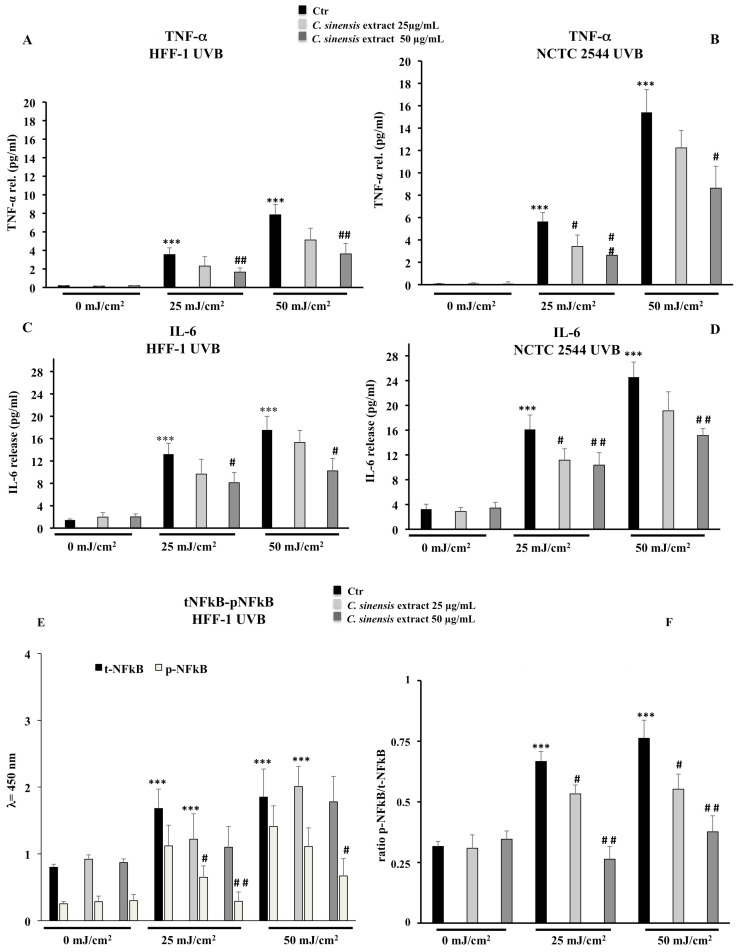
Effects of *C. sinensis* extract pretreatment (25 and 50 μg/mL) for 4 h on the release of TNF-alpha (**A**,**B**), IL-6 (**C**,**D**) and the activation of NFkB (**E**–**H**) after exposure to UVB (25 and 50 mJ/cm^2^) in HFF1 and NCT 2544 cells, respectively. The data are reported as the mean ± S.D. from triplicate experiments. * Significant vs. untreated control cells: *p* < 0.05; *** Significant vs. untreated control cells: *p* < 0.001; ^#^ Significant vs. untreated irradiated cells: *p* < 0.05; ^##^ Significant vs. untreated irradiated cells: *p* < 0.01; ^§^ Significant vs. treated irradiated *p* ≤ 0.05.

## Data Availability

Data were generated at the Department of Drug and Health Science, University of Catania. Data supporting the results of this study are available from the corresponding authors on request.

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
