# Peer review of "Phytocomplex of a Standardized Extract from Red Orange (Citrus sinensis L. Osbeck) against Photoaging"

_cells, 2022, doi:10.3390/cells11091447_

Round 1

Reviewer 1 Report

In this study, the effects red orange extract was investigated. It protected against negative effects of UV and reduced UV-induced ROS. Several molecular aspects were described.

The manuscript needs substantial improvement, also by considering the below list.

- All headings in Results section should be informative, e.g. instead of just “3.4.2. ROS assay” write in the headings what are the major findings, e.g. “3.4.2. ROS are reduced by …”. Applies to all headings in results.

- 3.1. Authors talk of Cell Viability and of cytotoxicity assays (which are not shown). These should be better shown in supplementary data. Authors should distinguish between cell viability and cytotoxicity.

- 3.2. Again, loss of cell viability, induced cytotoxicity and induced cell death are mixed up. Authors shall clearly state in Results text what has been done and which assay was used here (MTT). As MTT assay is also a cell proliferation assay and was applied at 24 h, it could be that mainly cell numbers are reduced. Another cell viability assay, e.g. calcein-AM staining should be applied to prove that protective activity of C. sinensis extract (Fig 2, Fig 3).

- Fig 2 and Fig 3 belong together in one figure.

- Most figures: Please explain the use of the sign for infinity. Possibly, it means irradiated. It is disturbing that it overlaps with the “g” for gram. Irradiation should be shown in a better way, e.g. as done in Fig 8a.

- Fig 4 legend. Please explain what you mean by “of three experimental determinations”. Are these 3 independent experiments or triplicates in one experiment?

- 3.3.2. The statement “This is probably due to the steric hindrance” needs further explanation.

- Fig 8 A/B, and other figures. Not good to use hatched bars. Also, the labeling is often too close to the X-axis.

- Fig 8G, 9D: Western blots should be shown together and in a separate figure

- All Western blots: Authors need to state, which size was obtained in experiments for the shown protein bands of the different factors. This should be determined according to a protein size marker. Necessary to understand that these are specific bands.

- Fig 9D Col1A1 and Elastin. The signals are completely unclear. They may be non-specific. Blots have to be repeated with another antibody. If there is no expression in these cells, a positive control for Col1A1 must be used and shown.

- Fig 9D MMP1. The contrast setting is particularly high. A standard setting should be used.

- Western blots statistics: Please state in the figures legends, how often the WB haave been repeated by using independently generated protein extracts. Two series of independent extracts should be minimum. Independent experiments should be shown in supplementary data.

- All figures – symbols for stat. significance: The ifferent symbols used are more confusing. Better use instead *, **, ***. Symbols must be better aligned, in the middle of bars and distant enough from the error bars.

- Discussion: It should be mentioned that many different herbal extracts can protect cells from ROS and UV effects. Examples should be listed. When explaining the effects seen here for this particular extract, authors should compare with the effects of other extracts. Thus, it is important to understand, whether the reported effects are specific for this extract or general.

Author Response

1 Reviewer

In this study, the effects red orange extract was investigated. It protected against negative effects of UV and reduced UV-induced ROS. Several molecular aspects were described. The manuscript needs substantial improvement, also by considering the below list.

We thank the reviewer for his/her kind comments and suggestions to improve the quality of our work. We have studied your comment carefully and have made the expected corrections, which we hope will meet your approval. Revised portions are highlighted in red in the manuscript.

  1. - All headings in Results section should be informative, e.g. instead of just “3.4.2. ROS assay” write in the headings what are the major findings, e.g. “3.4.2. ROS are reduced by …”. Applies to all headings in results.

A: The authors modified the headings in the Results section according to the reviewer's suggestion.

  1. - 3.1. Authors talk of Cell Viability and of cytotoxicity assays (which are not shown). These should be better shown in supplementary data. Authors should distinguish between cell viability and cytotoxicity.

A: The authors met the reviewer's suggestion

  1. - 3.2. Again, loss of cell viability, induced cytotoxicity and induced cell death are mixed up. Authors shall clearly state in Results text what has been done and which assay was used here (MTT). As MTT assay is also a cell proliferation assay and was applied at 24 h, it could be that mainly cell numbers are reduced. Another cell viability assay, e.g. calcein-AM staining should be applied to prove that protective activity of C. sinensis extract (Fig 2, Fig 3).

A: We thank the reviewer for pointing out this misunderstanding, we revised the whole manuscript only using “viability” as a parameter. We are very grateful for the reviewer’s suggestion; however, since our experiments included both untreated, but irradiated and un-irradiated, but treated cells, we are led to believe that the MTT assay, which detects viable cells with an active metabolism, might be used. We’ll consider this valuable suggestion for future experiments.   

- Fig 2 and Fig 3 belong together in one figure.

A: Considering the amount of results obtained by MTT assay on two different cell lines, it is difficult to include all graphs in one figure.  So for more clarity we prefer don't modify the figures.

  1. - Most figures: Please explain the use of the sign for infinity. Possibly, it means irradiated. It is disturbing that it overlaps with the “g” for gram. Irradiation should be shown in a better way, e.g. as done in Fig 8a.

 A: The infinity sign is a graphic error due to the different character sets of the computers; actually, the original sign is the Greek m (μ) that indicates "micro".

  1. - Fig 4 legend. Please explain what you mean by “of three experimental determinations”. Are these 3 independent experiments or triplicates in one experiment?

A: We thank the reviewer for pointing out this inattention. We revise the Fig. 4 legend: of three independent experiments in triplicate

  1. - 3.3.2. The statement “This is probably due to the steric hindrance” needs further explanation.

A: The authors met the reviewer's suggestion and added further explanation in the manuscript.

  1. - Fig 8 A/B, and other figures. Not good to use hatched bars. Also, the labeling is often too close to the X-axis.

A: The authors met the reviewer's suggestion

  1. - Fig 8G, 9D: Western blots should be shown together and in a separate figure

A: We appreciate the reviewer's suggestion but we have not fully understood it. However, we have consulted the journal guidelines as well as some published articles in Cells, and we believe that showing the western blot image separately does not meet the journal requirements. In addition, splitting the western blot images from the graphs would increase the number of figures and not make the visualization of the results clearer. However we increased the size of figures.

  1. - All Western blots: Authors need to state, which size was obtained in experiments for the shown protein bands of the different factors. This should be determined according to a protein size marker. Necessary to understand that these are specific bands.

A: We thank the reviewer for pointing out this inattention. We have modified Figures 8 and 9 by inserting the molecular weight of the marker and that of the analyzed proteins. In addition, we have also modified the caption of Figures 8 and 9.

  1. - Fig 9D Col1A1 and Elastin. The signals are completely unclear. They may be non-specific. Blots have to be repeated with another antibody. If there is no expression in these cells, a positive control for Col1A1 must be used and shown.

A: We thank the reviewer for his/her comment. In the NCTC 2544 keratinocytes, a model of skin epithelial-like cell line originating from normal human skin, the expression and the protein levels of Col1A1 and Elastin have been never detected in any experimental condition tested by us (ctrl, extract-treated cells, irradiated cells and so on). The WB data supports and confirms our PCR results obtained by using validated bioinformatic primers from Qiagen. In addition, Type I collagen is the predominant form of collagen in the human skin and is produced mainly by fibroblasts (Cutroneo, 2003) as well as elastin. Accordingly, since the anti-Col1A1 antibody by Cell Signaling correctly detected the protein in fibroblasts, cells physiologically responsible for the production of type I collagen, we believe that this can be considered a positive control and no further verification is required. Also, we removed the WB image of Col1A1 and Elastin for NCTC2544 from fig 9D.

  1. - Fig 9D MMP1. The contrast setting is particularly high. A standard setting should be used.

  A: According to the reviewer's suggestion the image has been replaced with a lower contrast one.

  1. - Western blots statistics: Please state in the figures legends, how often the WB haave been repeated by using independently generated protein extracts. Two series of independent extracts should be minimum. Independent experiments should be shown in supplementary data.

A: We thank the reviewer for his/her comment. Cells were seeded in a 12-well plate and treated for every experimental point in quadruplicate as described in the materials and methods section (experiments were repeated at least 2 times). Afterwards, cells treated in quadruplicate were collected in a single Eppendorf combining two replicates from the first experiment and two more from the second experiment in order to reduce possible differences in irradiation between the various wells of a plate; then the pellets were harvested with RIPA buffer containing Halt Protease and Phosphatase Inhibitor Single-Use Cocktail (Thermo Fisher). Lysates were separated by SDS-PAGE with 4–12% precast gel and transferred to nitrocellulose. We added the results of the second independent experiments in the supplementary file.

  1. - All figures – symbols for stat. significance: The different symbols used are more confusing. Better use instead *, **, ***. Symbols must be better aligned, in the middle of bars and distant enough from the error bars.

A: The authors met the reviewer's suggestion

  1. - Discussion: It should be mentioned that many different herbal extracts can protect cells from ROS and UV effects. Examples should be listed. When explaining the effects seen here for this particular extract, authors should compare with the effects of other extracts. Thus, it is important to understand, whether the reported effects are specific for this extract or general.

A: The authors revised the Discussion section according to the reviewer's suggestion.

Reviewer 2 Report

Dear authors,

The manuscript, you submitted, is very good and full of new data and perspectives about red oranges secondary metabolites and their future, potential use in medicine, cosmetic and nutrition. The work was also very well written and easy to read and understand. I have some minor remarks and objections, which will improve your text.

Title: Secondary metabolites from red orange - not oranges. Singular

Lines 36-40 divide into two sentences

Line 45 damages by production – to avoid repetition of due to in the line above

Line 45 reactive oxygen species (ROS) – ROS put into bracket

Line 85 in vitro – italic letters

Line 221-224 maybe to clarify better why did you use this particular concentration of extract (100  µg/mL)

Figure 2 add Ctr in the figure legend like you did in Figure 1

Replace value 15 j/cm2 on x-axis with 0 µg/mL because all values should be the same (extract concentrations)

Figure 3 the same observation as I Fig. 2 add Ctr in legend. 25 mJ/cm2 replace with 0 concentration of extract

Line 449 in vitro and in vivo italic letters

Author Response

The manuscript, you submitted, is very good and full of new data and perspectives about red oranges secondary metabolites and their future, potential use in medicine, cosmetic and nutrition. The work was also very well written and easy to read and understand. I have some minor remarks and objections, which will improve your text.

Dear Reviewer, thank you for your consideration of our manuscript and for the very valuable comments. We have read the comments carefully and revised the manuscript accordingly. Below we have included the comments and our point-by-point responses. Revised portions are highlighted in red in the manuscript.

  1. Title: Secondary metabolites from red orange - not oranges. Singular

A: The authors met the reviewer's suggestion

  1. Lines 36-40 divide into two sentences

A: The authors met the reviewer's suggestion

  1. Line 45 damages by production – to avoid repetition of due to in the line above

A: The authors met the reviewer's suggestion

  1. Line 45 reactive oxygen species (ROS) – ROS put into bracket

A: The authors met the reviewer's suggestion

  1. Line 85 in vitro – italic letters

A: The authors met the reviewer's suggestion

  1. Line 221-224 maybe to clarify better why did you use this particular concentration of extract (100  µg/mL)

            A: The authors used the dose of 100 µg/mL being the minimum dose to restore the viability of irradiated fibroblasts above 50%.

  1. Figure 2 add Ctr in the figure legend like you did in Figure 1

A: The authors met the reviewer's suggestion

  1. Replace value 15 j/cm2 on x-axis with 0 µg/mL because all values should be the same (extract concentrations)

A: The authors met the reviewer's suggestion

  1. Figure 3 the same observation as I Fig. 2 add Ctr in legend. 25 mJ/cm2 replace with 0 concentration of extract

A: The authors met the reviewer's suggestion

  1. Line 449 in vitro and in vivo italic letters

A: The authors met the reviewer's suggestion

Reviewer 3 Report

The manuscript in reference describes the study on a phytocomplex from three red orange varieties (Citrus sinensis (L.) Osbeck) in response to UVA-B-induced damages using an in vitro model and other biochemical measurements. The manuscript has relevant information and interesting results. However, some points are required to be addressed before further consideration.

  1. Detailed scrutiny is required to revise several grammar and stylistic issues throughout the manuscript.
  2. The title is not correct since the authors did not evaluate individual secondary metabolites rather than a standardized extract. In addition, the in vitro model and biochemical assays are not able to indicate that an extract is a useful tool. This idea is over scoped. Therefore, I strongly recommend modifying the title as “Standardized extract from Red Oranges (Citrus sinensis Osbeck) Against Photoagin”
  3. The abstract should involve some detailed results at the ending (lines 24-26).
  4. The introduction should be organized adequately since some sentences and passages are disconnected and wrongly written, involving short paragraphs without a clear order of ideas.
  5. Line 206: Data normal distribution must be verified prior to using these tests since ANOVA and Tukey are parametric tests. A parametric test cannot be used if data do not have a normal distribution. Be aware of that.
  6. Line 249: sinensis in italics. Be consistent throughout the manuscript.
  7. Line 413: is Vitamin C present in the extract? This information is not included in M&M section. Authors must include the content of Vitamin C.
  8. Discussion must be improved since it is laconically developed. The manuscript has numerous significant results that have not been adequately mentioned in this section.
  9. Conclusions are superficial and also very laconic. They should be rewritten, including more specific findings from a mechanistic point of view.

Author Response

The manuscript in reference describes the study on a phytocomplex from three red orange varieties (Citrus sinensis (L.) Osbeck) in response to UVA-B-induced damages using an in vitro model and other biochemical measurements. The manuscript has relevant information and interesting results. However, some points are required to be addressed before further consideration.

Dear Reviewer, thank you for your kind comments regarding our work. We have read the comments carefully and revised the manuscript accordingly. Below we have included the comments and our point-by-point responses. Revised portions are highlighted in red in the manuscript.

  1. Detailed scrutiny is required to revise several grammar and stylistic issues throughout the manuscript.

A: The complete manuscript was grammatically and stylistically revised according to the reviewer's suggestion

  1. The title is not correct since the authors did not evaluate individual secondary metabolites rather than a standardized extract. In addition, the in vitro model and biochemical assays are not able to indicate that an extract is a useful tool. This idea is over scoped. Therefore, I strongly recommend modifying the title as “Standardized extract from Red Oranges (Citrus sinensis Osbeck) Against Photoagin”

A: The title of the manuscript has been slightly changed to comply with the reviewer's request.

  1. The abstract should involve some detailed results at the ending (lines 24-26).

A: Considering the amount of results obtained on two different cell lines, it is difficult to include detailed results at the end (lines 24-26) of the abstract. However some indications have been added.

  1. The introduction should be organized adequately since some sentences and passages are disconnected and wrongly written, involving short paragraphs without a clear order of ideas.

A: The introduction section was revised according to the reviewer's request.

  1. Line 206: Data normal distribution must be verified prior to using these tests since ANOVA and Tukey are parametric tests. A parametric test cannot be used if data do not have a normal distribution. Be aware of that.

R: We thank the reviewer for this comment. We amended the text by clarifying how the statistical analysis was performed based on the data distribution.

  1. Line 249: sinensis in italics. Be consistent throughout the manuscript.

A: The word sinensis in italics has been revised throughout the manuscript.

  1. Line 413: is Vitamin C present in the extract? This information is not included in M&M section. Authors must include the content of Vitamin C.

A: We thank the reviewer for this comment. it was an oversight, we added the value of vitamin C in the M&M section

  1. Discussion must be improved since it is laconically developed. The manuscript has numerous significant results that have not been adequately mentioned in this section.

A: The discussion section was revised according to the reviewer's request.

  1. Conclusions are superficial and also very laconic. They should be rewritten, including more specific findings from a mechanistic point of view.

A: The conclusion section was revised according to the reviewer's request